# Intelligent Fault Diagnosis Method Based on Cross-Device Secondary Transfer Learning of Efficient Gated Recurrent Unit Network

**DOI:** 10.3390/s24134070

**Published:** 2024-06-22

**Authors:** Chaoquan Mo, Ke Huang

**Affiliations:** College of Mechanical & Electrical Engineering, Wenzhou University, Wenzhou 325035, China; 22451439021@stu.wzu.edu.cn

**Keywords:** mechanical equipment fault diagnosis, secondary transfer learning, cross-device, efficient gated recurrent unit network, continuous wavelet transform

## Abstract

In response to the issues of low model recognition accuracy and weak generalization in mechanical equipment fault diagnosis due to scarce data, this paper proposes an innovative solution, a cross-device secondary transfer-learning method based on EGRUN (efficient gated recurrent unit network). This method utilizes continuous wavelet transform (CWT) to transform source domain data into images. The EGRUN model is initially trained, and shallow layer weights are frozen. Subsequently, random overlapping sampling is applied to the target domain data to enhance data and perform secondary transfer learning. The experimental results demonstrate that this method not only significantly improves the model’s ability to learn fault features but also enhances its classification accuracy and generalization performance. Compared to current state-of-the-art algorithms, the model proposed in this study shows faster convergence speed, higher diagnostic accuracy, and superior robustness and generalization, providing an effective approach to address the challenges arising from scarce data and varying operating conditions in practical engineering scenarios.

## 1. Introduction

Rotating machinery is widely used in key engineering fields such as aerospace, automotive manufacturing, rail transportation, and wind power generation, and its operational performance directly impacts the overall safety of the equipment [1]. Condition monitoring and fault diagnosis are fundamental to ensuring the safe and stable operation of mechanical equipment [2]. Conducting the condition monitoring and diagnosis of mechanical equipment involves detecting, diagnosing, and predicting potential faults to prevent them from occurring, ensuring the reliability, continuity, and stability of mechanical operations, reducing economic losses and operating costs, and avoiding major accidents, which are of great engineering significance and practical necessity [3].

In recent years, the rapid development of next-generation artificial intelligence technologies, particularly deep-learning models, has attracted widespread attention from scholars worldwide and has been successfully applied in the intelligent fault diagnosis of mechanical equipment [4,5]. However, achieving excellent diagnostic performance with deep-learning models requires an ample supply of labeled data samples that are representative of the target distribution [6]. In engineering environments, available data are often scarce due to reasons such as cost, safety concerns, and varying operating conditions, making it difficult or even impossible to obtain data samples with labeled fault information. This scarcity of data leads to lower state recognition accuracy and weaker generalization capabilities of intelligent diagnostic models, while data-driven fault diagnosis models also require a significant amount of time and computational resources during training. Therefore, addressing data scarcity and variable operating conditions to achieve an intelligent diagnosis of health status has become a hot topic and challenge in the field of mechanical equipment fault diagnosis [7].

Data-driven intelligent fault diagnosis has become a hot topic of research in recent years. This diagnostic approach does not require the establishment of complex dynamic models of components or systems; instead, it relies on the use of large amounts of historical data to build and optimize models, making it very suitable for the intelligent fault diagnosis of complex machinery [8]. In recent decades, many data-driven fault diagnosis methods have been proposed, such as industrial process fault diagnosis based on wide convolutional neural networks with incremental learning abilities, the robust monitoring and fault isolation of nonlinear industrial processes using denoising autoencoders and elastic networks, and MoniNet for the concurrent analysis of time and space information for fault detection in the industrial processes [9]; dual-stream feature fusion convolutional neural networks for the parallel extraction of one-dimensional and two-dimensional fault features, and the adoption of feature fusion strategy for connection to achieve more reliable diagnostic results [10]; fine-tuning pre-trained models on ImageNet to improve fault classification performance in the target domain, with efficient and lightweight MobileNets [11]. Currently, data-driven diagnostic methods can generally be divided into three categories: those based on traditional machine learning, those based on deep learning, and those based on transfer learning [12].

Traditional machine-learning-based methods often rely on shallow network structures, limiting their ability to learn high-dimensional features, resulting in significant limitations in accuracy and generalization. Deep-learning-based methods, on the other hand, have made significant strides in the field of fault diagnosis due to their powerful automatic feature-learning capabilities [13,14,15,16]. However, deep-learning methods require a sufficient number of labeled fault samples for accurate classification results, and the training process of these models typically consumes a significant amount of time and computational resources. Transfer learning, a major breakthrough in artificial intelligence in recent years, aims to enhance the performance of models in the target domain by leveraging prior knowledge from the source domain [17].

Transfer learning (TL) can greatly relax the requirements for training samples (source domain data) to be abundant and must meet the mandatory requirement of being of the same distribution as the test samples (target domain data). TL has been successfully used in traditional pattern recognition fields such as image recognition, speech recognition, and text recognition [18]. In the field of mechanical intelligent diagnosis, transfer-learning methods have also attracted the attention of many researchers. Currently, transfer-learning methods can be broadly categorized as follows:

① Domain adaptation-based transfer-learning methods [19,20,21]: These methods aim to address the problem of distribution differences between the source domain and the target domain. They first reduce the differences between the source domain and the target domain and then use existing knowledge from the source domain to improve the learning performance on the target domain to complete the transfer under different operating conditions.

② Adversarial-based transfer-learning methods [22,23,24]: By introducing an adversarial network, these methods enable the model to simultaneously consider the distribution differences between the source domain and the target domain during the learning process. The training process of the model includes adversarial elements, allowing for more adaptive learning of significant features between the source domain and the target domain to improve performance on the target domain.

③ Pre-training-based transfer-learning methods [25,26,27]: Models are first pre-trained on a large-scale task and then transferred to the target task. The core idea of this method is to improve the performance of the model on specific target tasks by learning general features or representations from large-scale data. Its advantage lies in fully utilizing large-scale data for pre-training, thereby improving the model’s generalization ability and adaptability.

Transfer-learning methods have attracted the attention of researchers in the field of mechanical intelligent diagnosis and have achieved considerable results. However, these methods face some challenges. For example, domain adaptation transfer learning may have limited effectiveness for some complex domain differences, and simple domain adaptation methods may be insufficient. Some domain adaptation methods may require label information from the target domain, which can be costly to obtain in practical tasks. Adversarial-based transfer-learning training is often more complex than traditional methods, requiring more computational resources and training time. Adversarial methods like GANs are sometimes unstable, making the training process difficult. Pre-training transfer learning is usually performed on a large-scale dataset. However, available data in the field of mechanical equipment diagnosis are scarce, and pre-trained models often suffer from overfitting or underfitting, leading to lower diagnostic accuracy. Additionally, pre-trained models typically require a significant number of computational resources and time, making the training process expensive. Meanwhile, with the continuous improvement of automation and intelligence, there is a growing demand in engineering for high-precision and efficient models. To meet the requirements of engineering applications, there is an urgent need for a transfer learning model with high accuracy and short computational time.

In view of this, this paper proposes a method for intelligent fault diagnosis based on efficient gated recurrent unit network (EGRUN) cross-device secondary transfer learning. During the model construction phase, the combination of mobile convolutional structures (MBs) and Fused-MBConv (FMB) endows the model with excellent efficiency and accuracy. The gated recurrent unit (GRU) is introduced to further enhance the performance of the model, considering the time-series characteristics of bearing data. Continuous wavelet transform (CWT) can analyze signals at different time and frequency scales while capturing detailed information and transient impact signals, making it an excellent choice for processing mechanical equipment fault data. During the model training phase, data augmentation and second transfer learning are conducted using random overlapping sampling techniques, allowing the model to better learn mechanical equipment fault features and effectively improve the detection capabilities of the diagnostic network on target domain samples.

The main contributions of this method are as follows:A method for intelligent fault diagnosis based on EGRUN cross-device second transfer learning is proposed. The EGRUN model, consisting of core structures such as MB, FMB, and GRU, is constructed, with continuous wavelet time-frequency maps used as inputs, thereby improving the model’s feature extraction capabilities and diagnostic accuracy.Through data augmentation and secondary transfer-learning strategies using random overlapping sampling techniques, existing data are fully utilized, allowing the model to better learn mechanical equipment fault features, effectively improving the model’s fault recognition accuracy and generalization capabilities.

The remainder of this study is organized as follows: Section 2 explains the basic concepts of transfer learning; Section 3 elaborates on the proposed method; Section 4 evaluates the model’s performance on two different datasets through experiments; and Section 5 summarizes the work of this study and plans for future research.

## 2. Transfer-Learning Explanation

The general workflow of transfer learning involves utilizing data obtained from simulated mechanical equipment fault experiments in the laboratory to train a reliable transfer-learning model, which is then used to assess the operational status of mechanical equipment in real-world engineering scenarios. In specific applications, the data from the source domain typically consist of labeled mechanical equipment fault data obtained in laboratory experiments, while the data in the target domain usually lack labels. This is because the operational status of mechanical equipment in real-world engineering settings is highly uncertain. Therefore, current research on transfer learning primarily focuses on transferring knowledge between different experimental setups or different operating conditions of the same experimental setup [28].

In transfer learning, the domain being transferred from is referred to as the source domain (Ds), while the domain being learned is called the target domain (Dt). The Ds and Dt constitute two important domains in transfer learning. The data in the source domain are denoted as ${\left(x_{s}^{i},y_{s}^{i}\right)}_{i}^{n_{s}}$, where $n_{s}$ represents the number of samples in the source domain, $y_{s}^{i}$ represents the label corresponding to sample $x_{s}^{i}$, and $y_{s}^{i}\in \left\{1,2,3,...,k_{s}\right\}$ represents all the states among the samples in the source domain. The samples in the source domain come from the sample space $x^{s}$, and the labels come from the space $y^{s}$, indicating that $x_{s}^{i}\in X^{s}$, $y_{s}^{i}\in y^{s}$, and the data in the source domain follow the distribution $P_{s}\left(x,y\right)$. The data in the target domain are denoted as ${\left(x_{t}^{i},y_{t}^{i}\right)}_{i}^{n_{t}}$, where $n_{t}$ represents the number of samples in the target domain, $y_{t}^{i}$ represents the label of the sample $x_{t}^{i}$, and $y_{t}^{i}\in \{1,2,3,...,k_{t}\}$ represents all the states among the samples in the target domain. The samples in the target domain come from the sample space $x^{t}$, and the labels come from the space $y^{t}$, indicating that $x_{t}^{i}\in X^{t},\;y_{t}^{i}\in y^{t}$. The data in the target domain follow the distribution $P_{t}\left(x,y\right)$ [29], as illustrated in Figure 1.

## 3. EGRUN Diagnostic Scheme

### 3.1. Description of Network Structure

The EGRUN network structure and parameter settings are shown in Figure 2 and Table 1, respectively. Here, Conv3x3 denotes a standard 3x3 convolution + activation function (SiLU) + BN (Batch Normalization); Fused-MBConv module names followed by 1 or 4 indicate the expansion ratio, where k3x3 represents a kernel size of 3x3; MBConv module names followed by 4 or 6 indicate the expansion ratio, where k3x3 represents a kernel size of 3x3. SE0.25 indicates the use of the SE module, and 0.25 denotes that the number of nodes in the first fully connected layer of the SE module is 0.25 times the number of channels in the input feature matrix of the MBConv module; GRU stands for gate recurrent unit.

MBConv Structure

The MB structure mainly consists of a standard 1x1 convolution with BN and SiLU activation functions, which acts as a dimensionality-increasing component, followed by a depth-wise convolution with a kernel size of k × k and a stride of 1 or 2, along with BN and SiLU activation functions, significantly reducing the computational complexity and parameter count, where the value of k is primarily either 3 or 5; then, an SE module for channel weighting allocation; and a standard 1x1 convolution with BN to reduce dimensionality. Finally, a Dropout layer is added.

Residual structures first decrease dimensionality before increasing it, causing information loss in the intermediate operations, while inverted residual structures first increase dimensionality before decreasing it, resulting in minimal sparse feature loss. The MB structure is the fine-tuning of the inverted residual structure in the MobileNetV2 network model. Specific adjustments include (1) using different activation functions, replacing the Relu6 activation function with the Swish activation function, and (2) incorporating the channel attention mechanism into each MB by adding an SE structure after the depth-wise convolution layer to compute channel weights. Its structure is shown in Figure 3a.

2.Fused-MBConv Structure

The FMB structure modifies the MB structure by replacing a standard 1x1 convolution with the BN and SiLU activation functions and a depth-wise convolution with BN and SiLU activation functions, both with a kernel size of 1x1, with a single 3x3 standard convolution, and discarding the SE structure.

The FMB module has two forms: when the expansion ratio is 1, the convolution with the dimensionality-increasing effect is not used and is only applied in the first stage of the network; when the expansion ratio is not 1, a 3x3 convolution is used for the dimensionality-increasing effect and applied in the second and third stages. Its structure is shown in Figure 3b.

3.Stochastic Depth Dropout (SD-Dropout) Mechanism

The SD-Dropout layer in the network structure implements stochastic depth reduction, which means there is a possibility that only the shortcut branch remains, randomly dropping the main branch of the entire module, effectively reducing the depth of the network. Stochastic depth is not implemented using random dropout but rather by setting it as a linear function related to the number of inverted residual blocks. Assuming there is a total of L inverted residual blocks, the survival probability gradually decreases from P_0_ = 1 linearly to PL = 0.5 for the Lth inverted residual block. The formula is as follows:(1)$$pL=1-\frac{ı}{L}\left(1-pL\right)$$

The use of linear dropout is primarily because shallower layers extract low-level features, and later deeper layers are closely associated with these low-level features, so shallower layers should extract as many low-level features as possible and minimize dropout probability.

4.SE Channel Attention Mechanism

The SE attention mechanism adjusts the weight values of each channel in the shortcut branch based on different datasets and task environments, increasing or decreasing the importance of different channels during the training process. Its principle is shown in Figure 3c.

5.Gate Recurrent Unit Structure

The GRU [30] is a variant of recurrent neural networks (RNNs), similar to LSTM, which addresses issues such as the inability to maintain long-term memory and gradient vanishing during backpropagation in traditional RNNs. Compared to LSTM, GRU has a simpler internal architecture, has a faster computation speed, and is easier to train, greatly improving the training efficiency of the model. Adding a GRU layer helps capture the time dependence in sequential data, providing a unique advantage in tasks involving time-series information such as rotating machinery vibration data.

X_t_ is the input vector at time step t; H_t−1_ is the information from the previous time step t − 1; H_t_’ is the current memory content; Ht is the final memory content at the current time step; R_t_ is the reset gate; Z_t_ is the update gate. The reset gate determines the amount of information to forget from the previous time step, while the update gate measures the importance of past information and determines how much past information should be passed to the future or how much information from the previous time step to the current time step needs to be retained. Its structure is shown in Figure 4.

### 3.2. Continuous Wavelet Transform

The signals from rolling bearings fault are typically nonlinear and non-stationary, often exhibiting prominent amplitude-frequency characteristics that recur over time. Continuous wavelet transform (CWT) is a signal processing technique that enables the analysis of signals at different time and frequency scales, effectively capturing detailed information and transient impact signals. Therefore, CWT is well-suited for extracting fault features in rolling bearings, facilitating the analysis and processing of bearing health status.

CWT employs a function called a wavelet to perform local weighted analysis on the original signal. This wavelet function can vary in scale and frequency and is often referred to as the “mother wavelet” [31]. By convolving the mother wavelet function with the signal and then shifting and scaling it in time and frequency, CWT calculates the time-frequency representation of the signal. The mathematical expression of CWT is shown in Equation (2).
(2)$$CWT\left(a,\tau ,f\left(t\right)\right)=[f\left(t\right),\psi \left(a,\tau ,t\right)]=\frac{1}{\sqrt{a}}\int_{-\infty }^{+\infty }f\left(t\right)\psi^{*}\left(\frac{t-\tau }{a}\right)dt$$
where *f*(*t*) is the input signal; ψ is the mother wavelet function; $\psi^{*}$ is the complex conjugate of the mother wavelet function; *a* is the scale parameter controlling the width of the wavelet function; τ is the translation parameter controlling the position of the wavelet function in time.

Choosing the right wavelet basis function is essential for effectively extracting the fault features. Compared to other wavelet basis functions, the Morlet wavelet waveform is more similar to the temporal waveform of bearing operation [32]. The complex Morlet wavelet, or Cmorlet wavelet, is the complex form of the Morlet wavelet, offering better smoothness and adaptability, enabling more effective extraction of fault features from the original signal of rolling bearings. The mathematical expression of the Cmorlet wavelet is shown in Equation (3).
(3)$$\psi_{cmor}\left(t\right)=\frac{1}{\sqrt{\pi f_{b}}}e^{i2\pi f_{c}t}e^{-\frac{t^{2}}{f_{b}}}$$
where *t* is the time variable; $f_{c}$ is the center frequency; $f_{b}$ is the bandwidth parameter used to control the width of the wavelet function in the frequency domain.

At different scales, the wavelet function analyzes different frequency ranges of the signal, capturing local characteristics of the signal at different frequencies. The time-domain plot and CWT plot of different health conditions are shown in Figure 5.

### 3.3. Network Fault Diagnosis Process

The diagnostic process of the proposed EGRUN diagnostic scheme is illustrated in Figure 7, which can be summarized into three parts: preparing the sample set, training the model, and evaluating the model.

The diagnostic process of the proposed EGRUN diagnostic scheme can be summarized into three parts: preparing the sample set, training the model, and evaluating the model.

(1)Preparing the Sample Set

Vibration signal data under different operating conditions are obtained from mechanical equipment, and the raw data are sampled, divided into training and testing sample sets, and then corresponding CWT diagrams are generated. Sample extraction is generally carried out in two ways: non-overlapping and overlapping. Non-overlapping division may lead to the loss of some fault information due to fixed extraction positions, and in cases of short original signal lengths, fewer samples are generated. Conversely, overlapping division can, to some extent, avoid loss of sample information and produce more samples, making it a form of data augmentation. Although the use of overlapping division can generate more samples by increasing the overlap ratio, once the overlap ratio is determined, the position of each extracted sample in the original signal segment is fixed, still not fully utilizing the information from continuous signal segments.

In order to fully utilize the information of the original signal data and the randomness of the signal when a fault occurs, a new signal division method called random overlapping sampling is proposed in this paper. The term “random” refers to the random extraction positions of samples, while “overlapping” means that adjacent samples share the same part of the data points. By using random overlapping sampling to construct the sample set, different batches of samples are generated each time, as shown in Figure 6. With sufficient training, theoretically, signal segments from any position can be extracted, resulting in better data augmentation and enhancing the generalization ability of the diagnostic model.

(2)Training the Model

First Transfer-Learning Training: Initially, a sufficient sample of source domain data from the Case Western Reserve University bearing dataset is transformed into two-dimensional RGB images via continuous wavelet transform and inputted into the model for training. Then, the shallow layer weight parameters of the model are frozen to complete the first transfer learning. 

Second Transfer-Learning Training: The experimentally obtained dataset is used as the target domain, and data augmentation is performed on the data using random overlapping sampling techniques. Subsequently, the model is trained on the augmented dataset to adjust its ability to extract and learn deep features, completing the second transfer learning and allowing the model to fully learn and capture the fault characteristics of the data.

(3)Evaluating the Model

To verify the model’s learning ability and mastery of fault characteristics after transfer learning, various evaluation metrics such as the loss function value, diagnostic classification accuracy, confusion matrix, and t-distributed stochastic neighbor embedding (t-SNE) algorithm are used for the comprehensive evaluation of the model. The software environment used for fault diagnosis with the network model is PyCharm Community V2019.3.3_Tensorflow V2.13, and the hardware environment is Intel i7-12700H-2.3GHz_RAM-32GB.The diagnostic process is illustrated in Figure 7.

## 4. Fault Diagnosis Experimental Verification and Analysis

### 4.1. Data Description

(1)Dataset I

Dataset I is the Case Western Reserve University (CWRU)-bearing dataset, which is one of the widely used and internationally recognized public bearing datasets in the field of fault diagnosis. The dataset is generated through an experimental setup, as shown in Figure 8. The bearing models include the drive end bearing SKF-6205 and the fan end bearing SKF-6203. The faults of the bearings include single-point defects on the inner race (IRF), outer race (ORF), and rolling element (BF), with diameters of 0.007 inches, 0.014 inches, and 0.021 inches, respectively, created through electrical discharge machining. The sampling frequencies are 12 kHz and 48 kHz, and the vibration acceleration signals of the faulty bearings are collected using acceleration sensors. The dataset is divided into four operating conditions with motor loads of 0 HP, 1 HP, 2 HP, and 3 HP, corresponding to speeds of 1797 r/min, 1772 r/min, 1750 r/min, and 1730 r/min, respectively.

For the experiment, the signals from the drive end bearing are selected, with a sampling frequency of 12 kHz. There are four operating conditions, each containing 10 fault states. The labels range from 0 to 9, where label 9 represents normal bearing data, and labels 0 to 8 represent faulty bearing data. Each experiment uses a total of 1000 samples, with each sample consisting of 1024 data points. The data are divided into training set samples and testing set samples in a ratio of 8:2. The specific experimental data are shown in Table 2.

(2)Dataset II

Dataset II is collected from a rotating machinery fault simulation test platform, as shown in Figure 9. The experiment includes three different health states: bearing faults (rolling element faults), rotor bending faults, and normal states. Acceleration sensors are installed on the test platform to collect mechanical vibration signals with a sampling frequency of 20.48 kHz, under a load of 0 HP and a speed of 1500 r/min. Each experiment uses a total of 300 samples, with each sample consisting of 1024 data points. The data are divided into training set samples and testing set samples in a ratio of 8:2. The specific experimental data are shown in Table 3.

### 4.2. Description of Comparative Methods

Four classic deep-learning algorithms were compared:(1)ResNet50 Proposed by He et al. [33], ResNet50 is an important network in deep learning. It utilizes a residual learning framework, which simplifies the training of very deep networks by learning residual functions for layers rather than learning unknown functions.(2)MobileNetV3S Introduced by Howard et al. [34], MobileNetV3S is a lightweight convolutional neural network architecture designed to address real-time image recognition and processing on mobile devices.(3)EfficientNetV2S Proposed by Tan et al. [35], EfficientNetV2S is an efficient convolutional neural network architecture. It balances the depth, width, and resolution of the network using compound scaling methods to achieve better performance and efficiency.(4)Traditional Convolutional Neural Network (TCNN) is one of the classic networks in deep learning, playing a crucial role in the diversification and deepening development of subsequent deep-learning technologies, laying a solid foundation. The network structure is designed as a double convolutional layer network with specific parameters: the input size is (32,32); the number of convolutional kernels in the first convolutional layer is 8, with a size of (3,3) and a stride of 1; the number of convolutional kernels in the second convolutional layer is 16, with a size of (3,3) and a stride of 1; max-pooling is used for pooling, with a pooling kernel size of (2,2); and the final output layer performs the classification task using the Softmax function.

### 4.3. Diagnosis Results and Analysis

(1)Comparison of results on different transfer tasks.

The diagnosis results are shown in Figure 10. Here, “First Transfer Learning” refers to training using only the experimentally obtained dataset, while “Second Transfer Learning” refers to training the model first using the CWRU-bearing dataset and then using the experimentally obtained dataset for training, completing the second transfer learning.

Analysis of Transfer-Learning Results for Each Model

It can be observed that all five deep-transfer-learning diagnostic methods provide good classification results. Among the compared methods, TCNN has the lowest average classification accuracy (89.55%), followed by ResNet50 with an average classification accuracy of (95.00%), MobileNetV3S with an average classification accuracy of 97.50%, EfficientNetV2S with an average classification accuracy of 98.35%, and the proposed EGRUN with an average classification accuracy of 99.50%, which exhibits better classification performance compared to the other four comparison methods.

Furthermore, compared to single transfer learning after undergoing secondary transfer learning, the diagnostic accuracy of each method has been improved. TCNN improved by 1.9%, ResNet50 by 3.4%, MobileNetV3S by 1.6%, EfficientNetV2S by 0.9%, and the proposed method improved by 1%, achieving a diagnostic accuracy of 100%.

(2)Robustness Analysis

In actual industrial production, mechanical equipment typically operates under conditions that include various types of noise. To simulate real-world conditions, a certain level of Gaussian white noise is added to the original test data to evaluate the algorithm’s robustness to noise.

Gaussian noise is commonly used because it effectively describes the characteristics of noise in nature and many practical situations. Gaussian noise follows a Gaussian distribution, characterized by a mean of 0 and a standard deviation (σ) that determines the magnitude of the noise, ranging from [0, 1]. A higher value of σ indicates stronger noise and greater impact on the original data.

Data Noising Process

Using Dataset II as the base noisy data, Gaussian white noise with standard deviations of 0.2 and 0.4 is added to the original vibration signals to simulate various noise interferences present during actual bearing operations. The effects of adding Gaussian white noise to the data and CWT images are shown in Figure 11. The noisy signals are then used as training sets for the network, followed by testing to obtain the diagnostic accuracy results for different methods, as shown in the accuracy comparison in Figure 12.

Performance of Models under Noising Conditions

From the results in Figure 12, it can be seen that compared to the results of the original data without noise, the diagnostic accuracy of different model methods varies when different degrees of additional Gaussian white noise are added to the data. Among the five methods, EGRUN achieves the highest classification accuracy, with an average classification accuracy of 96.53%; followed by EfficientNetV2S, with an average classification accuracy of 94.60%; ResNet50 has an average classification accuracy of 89.70%; MobileNetV3S obtains poorer classification accuracy, with an average of 82.83%; and TCNN has the lowest classification accuracy, with an average of only 76.78%.

From Figure 11, it can be observed that the one-dimensional signal characteristics and two-dimensional CWT features with an added Gaussian noise of σ = 0.2 are compared with the original signal. The relevant discriminant information in the signal is interfered with and covered by noise, but most of the effective features are still retained. However, when Gaussian noise with σ = 0.4 is added, the relevant discriminant information in the signal is severely interfered with and covered by the noise, greatly increasing the difficulty of diagnosis for all methods.

Additionally, among the five methods, when Gaussian noise with σ = 0.2 is added, good results are still achieved, with average classification accuracies of 94.1%, 90%, 96.3%, and 98.3% for each model, respectively. When Gaussian noise with σ = 0.4 is added, the classification ability of all methods decreases significantly, especially for TCNN and MobileNetV3S. At this point, EGRUN and EfficientNetV2S still achieve relatively high average classification accuracies of 91.3% and 88.7%, respectively. MobileNetV3S has an average accuracy of only 60.0%, which aligns with the original design intention of this network. The average accuracy of TCNN is only 57.4%, indicating significant shortcomings in the ability of traditional convolutional neural networks to handle strong noise data.

The proposed method achieves higher classification accuracy among the five methods. The above analysis indicates that the proposed EGRUN exhibits better robustness against additional noise, demonstrating superior classification ability and generalization performance.

(3)Visualization Analysis (After Second Transfer Learning of the Model)

Analysis of Loss Function Values and Diagnostic Accuracy

During the experiment, the changes in the model’s loss function and diagnostic accuracy are shown in Figure 13. It can be seen that the loss function converges to 0 around 20 epochs, and the diagnostic accuracy also reaches its peak of 100% around 20 epochs. The consistency of the loss function values and diagnostic accuracy between the training and testing sets during the experiment indicates that the model did not suffer from overfitting during the parameter tuning process. Therefore, it can be concluded that the proposed EGRUN model exhibits excellent performance, achieving an accurate and stable fault diagnosis of mechanical equipment.

t-SNE Result Analysis

To further illustrate the effectiveness of EGRUN in feature extraction and diagnostic classification, the t-distributed stochastic neighbor embedding (t-SNE) algorithm is used to visualize the features of the input and output layers, demonstrating the model’s feature learning ability. As shown in Figure 14a, before classification, various types of data are mixed together in a chaotic manner, indicating a high level of confusion. However, as shown in Figure 14b, after undergoing feature extraction and learning by the EGRUN network, the intra-class compactness and inter-class separability of the features in the output layer have been significantly improved. The distances between different health states become clear and wide, while the data within the same class become more compact, and the confusion between different features disappears.

Confusion Matrix Result Analysis

Furthermore, to further demonstrate the classification performance of EGRUN, a confusion matrix analysis is conducted on one of the test results of the dataset, as shown in Figure 15. The proposed method accurately predicts all three health states of the mechanical equipment, resulting in a diagnostic accuracy of 100% in the final diagnosis results, consistent with the results in Figure 11. Therefore, it can be seen that the EGRUN model exhibits excellent fault diagnosis classification performance.

### 4.4. Comparison with State-of-the-Art Models in Terms of Diagnostic Accuracy

Diagnostic accuracy is one of the key indicators in the process of mechanical equipment fault diagnosis. The diagnostic accuracy of the ERGUN model was compared with that of other state-of-the-art algorithm models, as shown in Figure 16. These algorithm models include MMD-LMMD-TL [36], UTL-SOCNN-FBNM [37], MSSA [38], DIDA [39], and the models compared in the previous experiments.

From Figure 16, it can be seen that among the eight algorithms compared with the proposed model. The EfficientNetV2S model has the highest diagnostic accuracy, at 98.80%, while TCNN has the lowest diagnostic accuracy, at 90.50%. The diagnostic accuracy of the ERGUN model proposed in this paper is 9.5% higher than that of TCNN and still 1.2% higher than that of EfficientNetV2S.

Therefore, we can conclude that the proposed model can better extract and learn feature information, thereby achieving better fault diagnosis classification accuracy compared to other algorithms.

## 5. Conclusions

This study has successfully developed an intelligent fault diagnosis method based on EGRUN for cross-device secondary transfer learning, effectively addressing the challenge of mechanical equipment fault diagnosis caused by data scarcity. The main conclusions of this study are summarized as follows:(1)The EGRUN model, which combines the MBConv, Fused-MBConv, and GRU core structures, was designed. By using CWT images as input, the model’s feature extraction and diagnostic accuracy were significantly enhanced.(2)Data augmentation and secondary transfer learning were conducted using random overlapping sampling techniques, enabling the model to learn fault features more effectively and significantly improving the diagnostic performance on target domain samples.(3)Experimental validation demonstrated that compared to eight advanced deep-transfer-learning algorithms, the proposed method exhibited superior feature learning capability, fault classification accuracy, and generalization performance. With a diagnostic accuracy of 100%, it demonstrated the potential for practical applications.

## Figures and Tables

**Figure 1 sensors-24-04070-f001:**
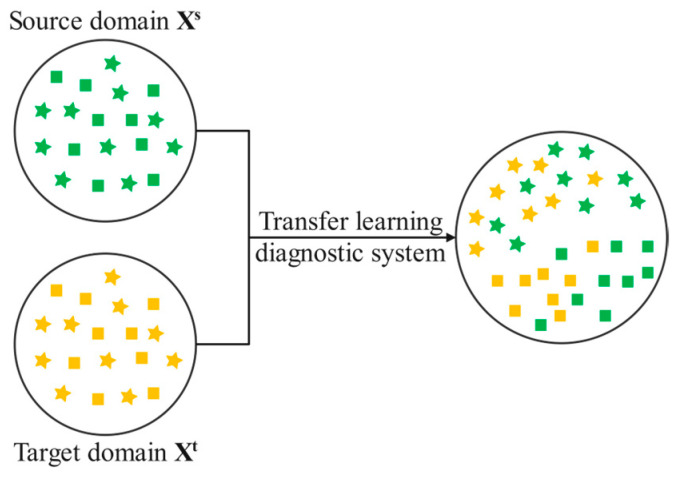
Schematic diagram of X^s^ → X^t^ transfer learning.

**Figure 2 sensors-24-04070-f002:**
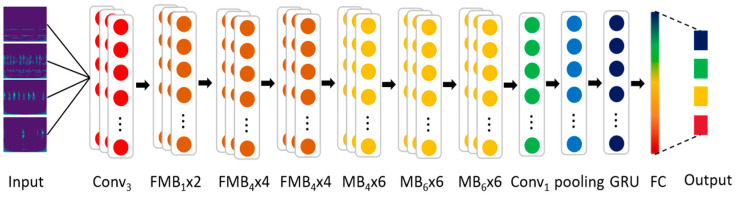
Flowchart of the EGRUN structure.

**Figure 3 sensors-24-04070-f003:**
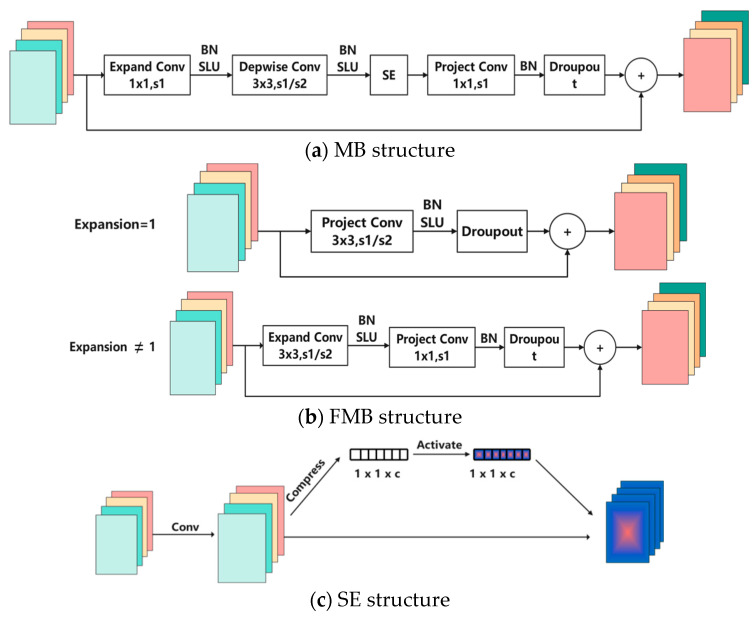
MB, FMB and SE structure.

**Figure 4 sensors-24-04070-f004:**
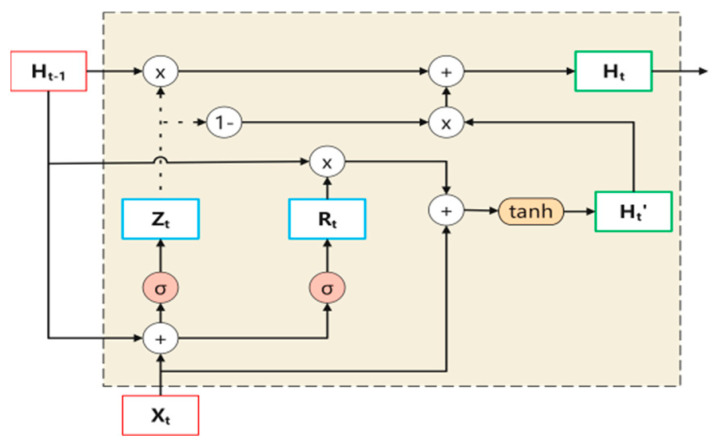
Basic structure of GRU.

**Figure 5 sensors-24-04070-f005:**
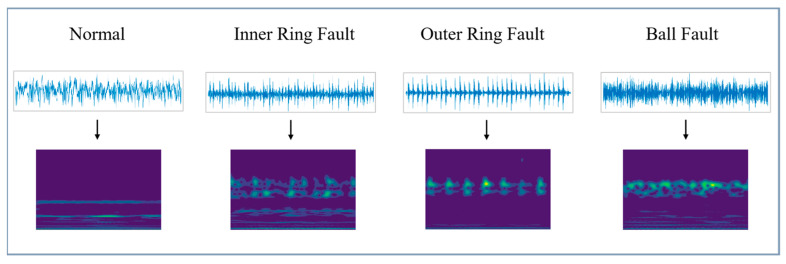
Time-domain plots and CWT plots of different health conditions.

**Figure 6 sensors-24-04070-f006:**
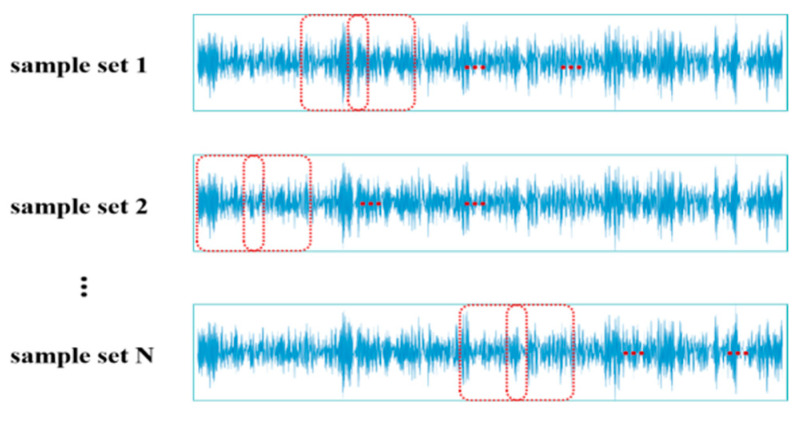
The concept of random overlapping cropping for the sample set.

**Figure 7 sensors-24-04070-f007:**
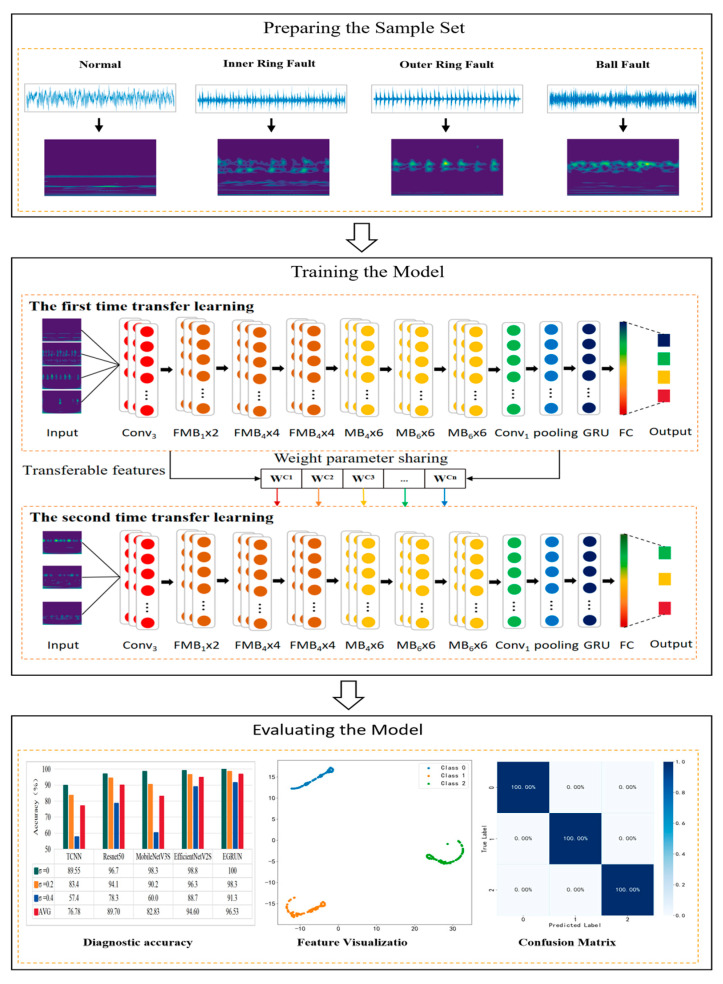
Diagnostic process of the proposed model.

**Figure 8 sensors-24-04070-f008:**
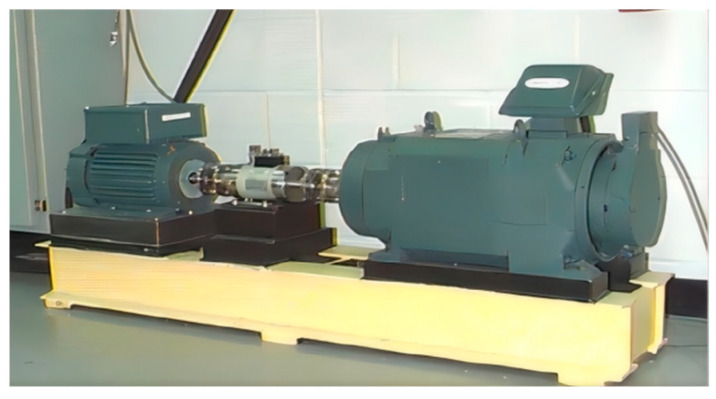
CWRU-bearing fault diagnosis experimental platform.

**Figure 9 sensors-24-04070-f009:**
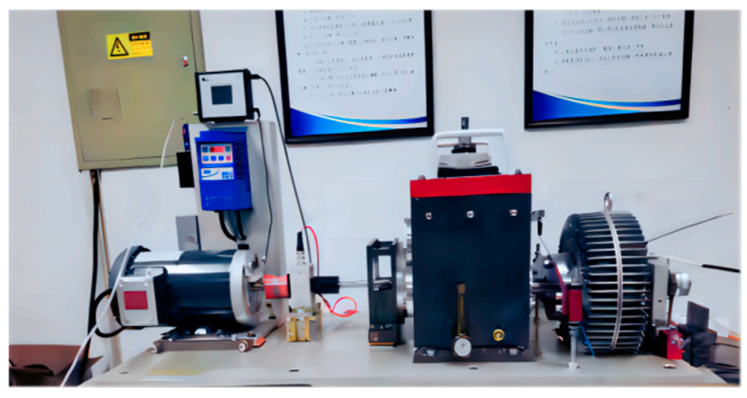
Simulation test platform for mechanical equipment faults.

**Figure 10 sensors-24-04070-f010:**
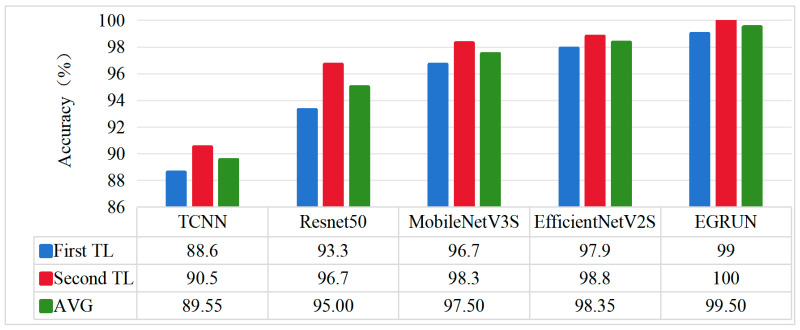
The classification results of different methods.

**Figure 11 sensors-24-04070-f011:**
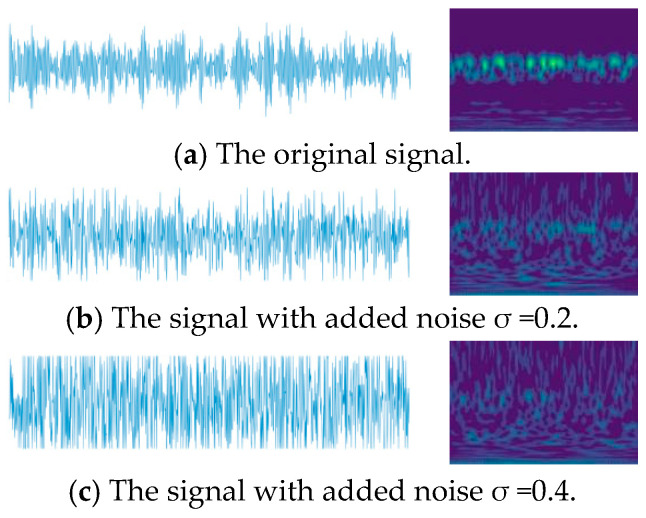
Data and CWT images after adding Gaussian noise.

**Figure 12 sensors-24-04070-f012:**
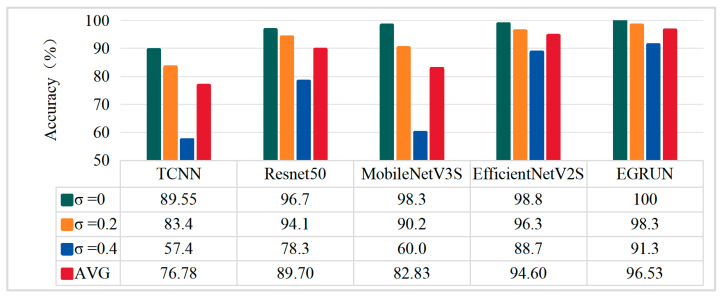
Diagnostic accuracy of various models under different noise levels.

**Figure 13 sensors-24-04070-f013:**
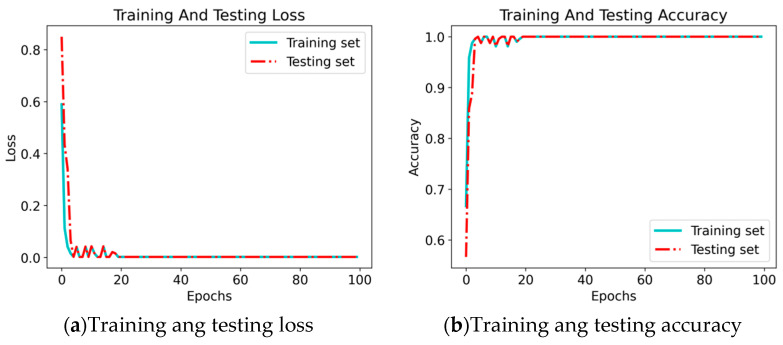
Loss function values and diagnostic accuracy.

**Figure 14 sensors-24-04070-f014:**
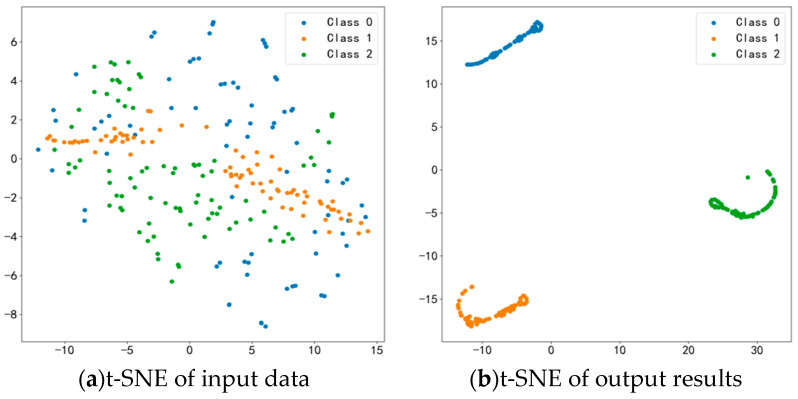
Feature visualization.

**Figure 15 sensors-24-04070-f015:**
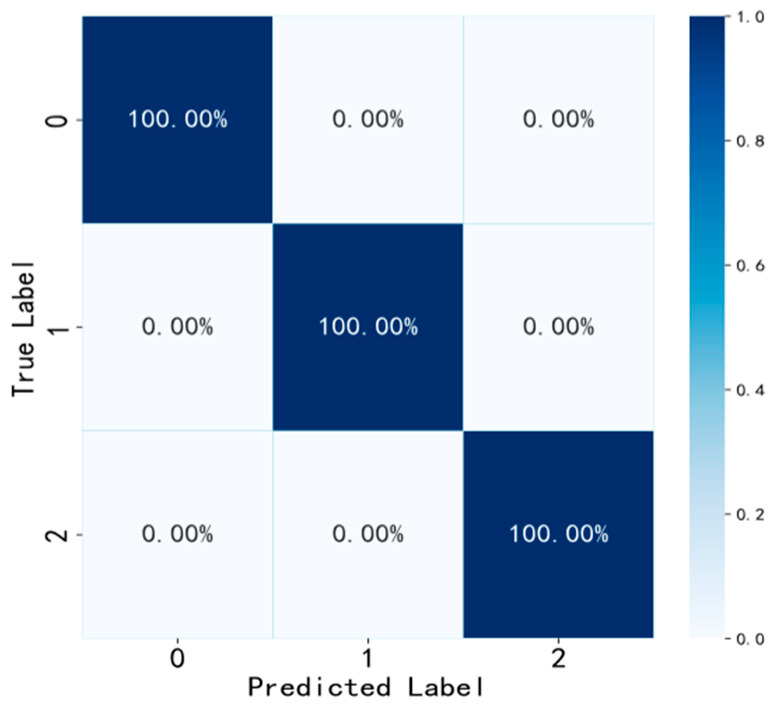
Confusion matrix.

**Figure 16 sensors-24-04070-f016:**
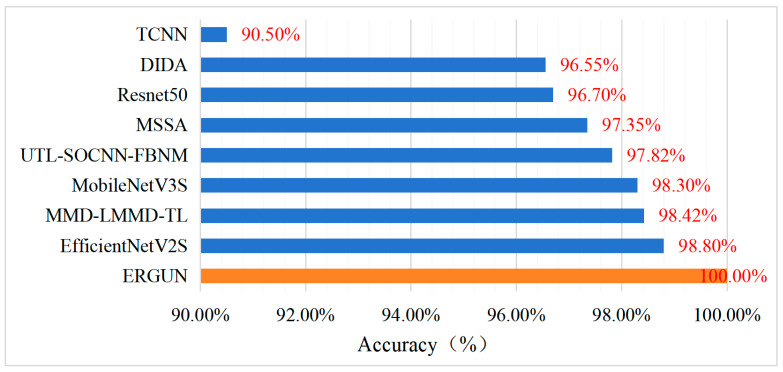
Comparison of diagnostic accuracy with other state-of-the-art algorithms.

**Table 1 sensors-24-04070-t001:** Network structure parameters of the proposed methods.

Stage	Operator	Layers	Stride	Channels
1	Input	1	-	3
2	Conv3x3	1	2	24
3	Fused-MBConv1, k3x3	2	1	24
4	Fused-MBConv4, k3x3	4	2	48
5	Fused-MBConv4, k3x3	4	2	64
6	MBConv4, k3x3, SE0.25	6	2	128
7	MBConv6, k3x3, SE0.25	9	1	160
8	MBConv6, k3x3, SE0.25	15	2	256
9	Conv1x1 & Pooling	1	-	1280
10	GRU	1	-	128
11	Flatten & FC	1	-	128
12	Output	1	-	10/3

**Table 2 sensors-24-04070-t002:** Description of bearing experiment data.

Label	Status	Fault Sizes (mm)	Training Set	Testing Set	Sample Lengths
0	B007	0.1778	80	20	1024
1	B014	0.3556	80	20	1024
2	B021	0.5334	80	20	1024
3	IR007	0.1778	80	20	1024
4	IR014	0.3556	80	20	1024
5	IR021	0.5334	80	20	1024
6	OR007	0.1778	80	20	1024
7	OR014	0.3556	80	20	1024
8	OR021	0.5334	80	20	1024
9	Normal	—	80	20	1024

**Table 3 sensors-24-04070-t003:** Description of bearing experiment data.

Label	Load/RPM	Status	Training Set	Testing Set
0	0 HP, 1500 r/min	Bearing fault	80	20
1	0 HP, 1500 r/min	Rotor fault	80	20
2	0 HP, 1500 r/min	Normal	80	20

## Data Availability

Data can be made fully available upon request.

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
