# Peer review of "Intelligent Fault Diagnosis Method Based on Cross-Device Secondary Transfer Learning of Efficient Gated Recurrent Unit Network"

_sensors, 2024, doi:10.3390/s24134070_

Round 1

Reviewer 1 Report

Comments and Suggestions for Authors

The study introduces an intelligent fault diagnosis method using EGRUN and cross-device secondary transfer learning, which addresses data scarcity and variable operating conditions, outperforming other methods in diagnostic accuracy and generalization, and showing strong potential for practical engineering applications. Overall, this study presents an intelligent fault diagnosis method using an Efficient Gated Recurrent Unit Network (EGRUN) and cross-device secondary transfer learning, addressing data scarcity and variable conditions in mechanical equipment. By incorporating Mobile Convolutional Structures, Fused-MBConv, and Gated Recurrent Units, and employing data augmentation and two-stage transfer learning, the method outperforms ResNet50, MobileNetV3S, and EfficientNetV2S in diagnostic accuracy, robustness, and generalization, providing technology in fault diagnosis system.

However, there are still some improvements that need to be made and additional corrections required in the manuscript:

* Images should not be stretched in one direction only; they need to be proportionally scaled in both dimensions.

* The paper should provide more details on the specific operating conditions (load, speed, etc.) of the mechanical equipment used in the experiments.

* The paper could benefit from a comparison of the proposed method's performance against traditional machine learning techniques or simpler deep learning models.

* The author should investigate the impact of using different wavelet bases or mother wavelet functions for the continuous wavelet transform on the feature extraction and classification performance.

* The paper could discuss more about the scalability of the proposed method to handle larger datasets or higher-dimensional input data.

* Section 4 (Fault Diagnosis Experimental Verification and Analysis): The structure is quite dense, with multiple experimental results, visualizations, and comparisons presented. Using subheadings and bullet points to organize key findings and performance metrics for each experiment could improve readability and clarity.

* Visualization of Continuous Wavelet Transform Images: The visualization in Figure 3 could be improved by highlighting the key features or patterns discussed in the text, making it easier for readers to correlate the visual representation with the analysis.

* The paper should discuss the potential for incorporating additional sensor modalities or data sources (e.g., temperature, vibration, acoustic emissions) into the proposed fault diagnosis framework.

* The abstract and conclusion need to be rewritten concisely, focusing on the methods and achieved results, as well as the main contributions.

Comments on the Quality of English Language

English style and usage is fine.

Author Response

The following are our responses to each comment and suggestion. Please see the attachment.

Reviewer 2 Report

Comments and Suggestions for Authors

In this manuscript, an intelligent fault diagnosis method is proposed based on efficient gated unit network cross-device secondary transfer learning. Experimental results on a public bearing dataset and a rotating machinery fault simulation test platform illustrate the performance of the proposed method. Some comments about this study are presented below

1)     It is suggested to delete the third contribution since every method has been illustrated using experimental results.

2)     A schematic diagram should be provided for better illustrating the transfer learning in Section II. Besides, some related literatures should be provided to support your conclusion in this section.

3)     Many data driven fault diagnosis methods have been proposed in recent decades. The reviewers should mention them in the introduction to make the background more comprehensive. Such as: broad convolutional neural network based industrial process fault diagnosis with incremental learning capability, robust monitoring and fault isolation of nonlinear industrial processes using denoising autoencoder and elastic net, MoniNet with concurrent analytics of temporal and spatial information for fault detection in industrial processes.

4)     A flow chart about the proposed method should be provided to make a better clarification. It is quite confused in the methodology part.

5)     What is the difference between the proposed method and the existing diagnosis methods?

6)     How to select the model parameters of the proposed method. It is better to analyze the effect of the model parameters.

7)     The model parameters of the selected comparison methods should be provided to make a comprehensive comparison.

Comments on the Quality of English Language

The authors are suggested to carefully checked the manuscript to avoid grammer and spelling mistakes.

Author Response

(The authors gave the same response as above.)

Reviewer 3 Report

Comments and Suggestions for Authors

The authors present a method for intelligent fault diagnosis. In the literature review section well-justified-by references to the state-of-art published works- stated the need for "a transfer learning model with high accuracy and short computational time". The authors propose the method "Efficient Gated Recurrent Unit Network (EGRUN) Cross-Devise Second\ry Transfer Learning. The proposed algorithm is explained sufficiently well. It would be useful to add some computational mechanics motivation - interpretation to the various modulus in the algorithms depicted in Fig. 1. The information in the text seems to be oriented to informatics educated people whereas the application here is for a complicated mechanical system.

The presented computations on the public available databases (source and target domains) of vibration datasets of actual mechanical systems support the claim that the proposed fault diagnostic method has features to justify the characterization as "intelligent". 

It seems that coined term "intelligent" is the result of transfer learning and data augmentation by random overlapping and the architecture of the proposed algorithm.

Figure 14 tells us that ERGUN has a 100% accuracy. I believe this cannot be generalized. It is only for the landscape of the available datasets. The landscape might have regions where even ERGUN does not score 100% accuracy. 

The article is well-written and its finding will benefit interested researchers and engineers. Figure 5 needs more explanation in the caption. Lines 169-176 should be revised to explain better the stages in Table 2.

Author Response

(The authors gave the same response as above.)

Round 2

Reviewer 1 Report

Comments and Suggestions for Authors

The revised version is well-prepared and ready for publication.

Author Response

Dear Reviewer,

Thank you very much for your attention and time. We appreciate once again your recognition of our work. Wishing you a pleasant day!

Yours sincerely,

Ke Huang  

18 June 2024  

Reviewer 2 Report

Comments and Suggestions for Authors

Mos of the questions I raised have been addressed. Before the article is accepted, the authors should add the related references of the newly added methods on Lines 50-59 on Page 2.

Comments on the Quality of English Language

The authors should carefully checked the article to avoid grammar and spelling mistakes.

Author Response

Dear Reviewer,

1) Mos of the questions I raised have been addressed. Before the article is accepted, the authors should add the related references of the newly added methods on Lines 50-59 on Page 2.

Response:

Thank you for your comments and suggestions. In order to better illustrate the relevant methods, based on your advice, we have re-added relevant references in the method added on page 2. The revised content is as follows:

In recent decades, many data-driven fault diagnosis methods have been proposed, such as: industrial process fault diagnosis based on wide convolutional neural networks with incremental learning ability, robust monitoring and fault isolation of nonlinear industrial processes using denoising autoencoders and elastic networks, MoniNet for concurrent analysis of time and space information for fault detection in industrial processes[10]; dual-stream feature fusion convolutional neural networks for parallel extraction of one-dimensional and two-dimensional fault features, and adoption of feature fusion strategy for connection to achieve more reliable diagnostic results[11]; fine-tuning pre-trained models on ImageNet to improve fault classification performance in the target domain, with efficient and lightweight MobileNets[12].

[10]Yu W, Zhao C, Huang B. MoniNet with concurrent analytics of temporal and spatial information for fault detection in industrial processes[J]. IEEE Transactions on Cybernetics, 2021, 52(8): 8340-8351.

[11]Xue F, Zhang W, Xue F, et al. A novel intelligent fault diagnosis method of rolling bearing based on two-stream feature fusion convolutional neural network[J]. Measurement, 2021, 176: 109226.

[12]Yu W, Lv P. An end-to-end intelligent fault diagnosis application for rolling bearing based on MobileNet[J]. IEEE Access, 2021, 9: 41925-41933.

Yours sincerely,

Ke Huang  

18 June 2024  
